# Electrical and Optical Properties of CaTi_1−y_Fe_y_O_3−δ_ Perovskite Films as Interlayers for Optoelectronic Applications

**DOI:** 10.3390/ma15196533

**Published:** 2022-09-21

**Authors:** Ceren Yildirim, Fabien Devoize, Pierre-Marie Geffroy, Frédéric Dumas-Bouchiat, Johann Bouclé, Sylvain Vedraine

**Affiliations:** 1IRCER, CNRS, Université de Limoges, CEC, 87068 Limoges, France; 2XLIM, CNRS, Université de Limoges, 87060 Limoges, France

**Keywords:** perovskite, optoelectronics, KPFM, conductivity, absorption, solar cell

## Abstract

CaTi_1−y_Fe_y_ O_3−δ_ perovskite oxide films are promising candidate materials for p-type interlayers of third generation solar cells or light-emitting devices. The impact of atomic Ti substitutions by Fe on electrical and optical properties of CaTi_0.5_Fe_0.5_O_3−δ_ perovskite films have been studied. The best compromise between a high transmission coefficient and the suitable electrical conductivity is obtained for a specific atomic composition of Ca (1) Ti (0.5) Fe (0.5) O (3−δ) perovskite films. This paper shows that CaTi_1−y_Fe_y_O_3−δ_ perovskite oxides can be integrated as p-type interfacial layers of optoelectronic devices through their work functions, electrical, and optical properties.

## 1. Introduction

Semi-conducting perovskite oxides show promise for optoelectronic applications due to a broad spectrum of electrical and optical properties [1,2,3,4]. The electrical conductivity, band gaps, transparency, and work function could be engineered with the nature of the cation substitution of the perovskite structure [5]. Recent works have identified CaTi_1−y_ Fe_y_O_3−δ_ perovskite oxides as promising candidate materials for interlayer for electronic applications [6].

Moreover, CaTi_1−y_Fe_y_O_3−δ_ perovskite oxides show a wide variety of electrical and likely optical properties, such as a high transparency in the visible range in relation with Ti substitution by Fe. This specific substitution leads to large structural and electrical conductivity variations [7,8,9]. CaTi_1−y_Fe_y_O_3−δ_ perovskite oxides are usually studied for the ionic and electronic conductivities at high temperatures for oxygen transport membrane application [9,10,11]. Curiously, quite few data on both electrical and optical properties at room temperature are reported in the literature, and this perovskite oxide series has never been investigated in detail for photonic applications as an interlayer.

This paper is focused on the development of oxides based on CaTi_1−y_Fe_y_O_3−δ_ perovskite materials. For this work, a particular attention is given to the impact of the Ti substitution ratio by Fe on the optical and electrical properties of perovskite oxide thin films developed by pulsed laser deposition (PLD) for three configurations.

## 2. Materials and Methods

### 2.1. Synthesis of CaTi_1−y_Fe_y_O_3−δ_ Perovskite Targets

CaTi_1−y_Fe_y_O_3−δ_ perovskite powders were synthesized using a solid-solid route. High-purity oxide and carbonate precursors, including CaCO_3_ (99.99%, Sigma-Aldrich, Saint-Louis, MO, USA), Fe_2_O_3_ (99.50%, Alfa Aesar, Haverhill, MA, USA), and TiO_2_ (99.90%, Alfa Aesar, Haverhill, MA, USA), were mixed by attrition milling using 0.9–1.2 mm zirconia balls in an ethanol media (Sigma Aldrich, Saint-Louis, MO, USA), (800 rpm for 1 h) and the powders were calcined for 8 h at 1000 °C, and then the phase purity was confirmed with X-ray diffraction (Siemens D5000, Cu-Kα, Munich, Germany). After the synthesis, the chemical stoichiometry of CTF powders had less than 5% variation from the chemical composition expected, confirmed by Inductively coupled plasma atomic emission spectroscopy (ICP-AES). An attrition milling of the calcined powders (1000 rpm) reached a distribution of monomodal grain size centered at about D_50_ = 1–2 µm. Water of polyvinyl alcohol and polyethylene glycol (Sigma Aldrich, Saint-Louis, MI, USA), 200 were added then mixed into the powder respecting the following ratio (50, 1.5, 1.5%wt.), and the powder was dried at 90 °C for 10 h at the next phase.

Afterwards, the powders were pressed under 200 MPa at 25 °C and sintered to obtain green pellets of 24–25 mm in diameter and 1 mm in thickness. The pellets were sintered at 1400 °C for 6 h under air, as reported in Table 1. The density of the sintered pellets was measured using the Archimedes’ method. Three oxides corresponding to different Ca substitution ratios by Fe were studied in this work: CaTi_0.7_Fe_0.3_O_3−δ_ (CTF73), CaTi_0.5_Fe_0.5_O_3−δ_ (CTF55), and CaTi_0.3_Fe_0.7_O_3−δ_ (CTF37), as reported in Figure 1.

Figure 1 shows the phase diagram of the CaTiO_3_–CaFeO_2.5_ system. It shows that the structure of CaTi_1−y_FeyO_3−δ_ oxides depended on the Ti substitution ratio by Fe (y) in the perovskite structure. Then, we distinguished different structures of the oxide at room temperature in relation with the ratio of Fe in perovskite structure as shown below:

(i) ‘Disordered region’ (x = 0–0.1) where there was no long-range ordering of oxygen vacancies in the perovskite structure.

(ii) ‘Partially ordered region’ (x = 0.1–0.5) where the oxygen vacancies were partially ordered in the perovskite structure, which corresponds to CTF73 material.

(iii) ‘TOO + TOOO region’ (x = 0.5–0.65) where there was regular sequence of one tetrahedral (T) and two or three octahedral layers (O), which corresponds to CTF55 material.

(iv) ‘TO + TOO region’ (x > 0.65) where regular sequence of one tetrahedral (T) and one or two octahedral layers (O), which coincided to CTF37 material.

Figure 2 confirms the evolution of perovskite structure in relation with the Ti substitution ratio by Fe. As disclosed on the phase diagram of the CaTiO_3_–CaFeO_2.5_ system in Figure 1, the X-ray diffraction pattern of CaTi_0.7_Fe_0.3_O_3−δ_ (CTF73) corresponded to the cubic phase, CaTi_0.5_Fe_0.5_O_3−δ_ (CTF55) corresponded to the cubic phase with the TO sequence, and CaTi_0.3_Fe_0.7_O_3−δ_ (CTF37) corresponded to the TO and TOO sequence layers.

### 2.2. Elaboration of CaTi_1−y_Fe_y_O_3−δ_ Perovskite Films

Pulsed Laser Deposition (KrF—248 nm, 10 Hz) in an ultra-high vacuum chamber (VINCI Technologies, France, 10^−8^ mbar) had been used to prepare the perovskite films. At this UV-wavelength regime, photons are highly energetic (5 eV). Consequently, the deposition process was well known to be very congruent. Targets of the desired composition were sprayed at a fluency of around 3 J·cm^−2^ under dynamic oxygen pressure (pO_2_ = 0.3 mbar). During deposition, fluorine-doped tin oxide- (FTO) coated glass slide substrates were heated at 400 °C and used as substrates. Deposition rates measured at around 0.20 nm·s^−1^, for a target–substrate distance of 5 cm, a standard value observed for oxide materials. To improve perovskite crystallization, samples received post-annealing at 550 °C under N_2_ pressure for 1 h.

The crystallographic phase of thin films was determined by (θ, 2θ) X-ray diffraction (XRD) (Siemens D5000, Cu-Kα, Munich, Germany). Notably, the X-ray diffraction patterns confirmed the presence of a perovskite phase, as identified by the sign of star marks above the peak in Figure 3 [13,14]. The CaTi_1−x_Fe_x_O_3−δ_ coating obtained by PLD showed a less intensive peak with a lower crystallinity degree related to thin films thickness (100 nm). However, it can be clearly stated that a significant increase in peak intensity was observed following the post-annealing process at 550 °C. In addition, the crystals of the CTF55 thin film emerged on the pattern after annealing.

The cross-section SEM (Scanning Electron Microscope) image (Figure 4a) (Jeol, Tokyo, Japan) shows a dense homogenous columnar microstructure of 100 nm thin CTF55 film deposited on FTO substrate (200 nm thick). The anisotropic (directional) PLD process is well known to lead to columnar growth, especially for oxide materials. The columnar microstructure of the CTF55 film is a consequence of an out-of-plane homogenous grain growth along an axis perpendicular to the substrate surface. Then, the relative intensity of the peaks on the XRD patterns of CFT55 film (Figure 3) was largely affected by the anisotropic microstructure of film in comparison with one of pellet (Figure 2). As usual, microstructure and crystal intrinsic properties manage the macroscopic layer properties. Microstructure quality appears to be essential for the development of the materials of interest. Several SEM cross-section images (not shown here) confirm the CTF55 film homogeneity. The top view SEM image confirms the high density of the deposited film as seen in Figure 4b and the small grain size dimensions in the x–y substrate plane (10–20 nm).

The homogeneous and dense micro-structure material directly influences the path that conductors take in the oxide materials. Therefore, these properties have a positive impact in terms of the electrical and optical performance of the thin films.

### 2.3. Electrical Characterizations

The dense perovskite pellets (24–25 mm in diameter and 1 mm of thickness) used as a target for PLD were electrically characterize using an impedance spectroscope. The two faces of the target were coated with platinum electrodes via platinum paste (Pt paste, Ferro, CDS electronique, Saint-Dizier Cedex, France). The samples were heated up to 1000 °C in air to obtain cohesive and porous platinum electrodes. The impedance measurements were performed between 20 °C and 150–350 °C using frequencies ranging from 1 Hz to 1 GHz with a signal amplitude of 300 mV (Solatron 1260). The purpose of this measurement is to obtain results about the physical properties and internal structure of CTF thin films.

The electrical conductivity characterization of CTF thin films deposited by PLD was performed using a gold microelectrode (0.4 × 0.4 mm^2^) obtained via photolithography method on the top of thin film and the metallic tip in contact with the FTO surface, as reported on Figure 5. A Keithley 2400 is used for the electrical measurement based to the two-point probe method. For these electrical characterizations, the thicknesses of CTF films were thicker, close to 200 nm in thickness, to improve the measurement accuracy.

### 2.4. Optical Characterizations

Optical reflection and transmission of thin perovskite films of 150–200 nm in thickness on C-type sapphire substrate (Neyco/BT Electronics) were carried out using an Cary 300 reflectometer (AGILENT Santa Clara, CA, USA). Absorbance α was deduced from transmission T (Equation (1)) using a thickness d. The Tauc plot method [15] (Equation (2)) links the incident photon energy hυ, the optical gap Eg, the absorbance, and the constants r (1/2 or 2 for the direct and indirect transition band gaps, respectively) and B (corresponding to the band tailing parameter). The *x*-axis intersection point of its linear fit gives an estimate of the optical band gap Eg.
(1)α=−(1/d)log(T)
(2)$$\alpha h\upsilon =B^{2}{\left(h\upsilon -E_{g}\right)}^{r}$$

### 2.5. Surface Potential Measurements

The Nano-Observer AFM (Atomic Force Microscope) from CSI^©^ (Les ULIS, France) including HD Kelvin Probe Force Microscopy (HD-KPFM) was used to measure the local contact potential difference $V_{\mathrm{measured}}$ between a conducting atomic force microscopy tip and the selected sample. Oxide layers on the glass confirmed the presence of flat surfaces with root mean squared (RMS) roughness measured at about 10 ± 3 nm for all CTF samples. Roughness was increased to 12 nm (±1 nm) for all CTF layers 100 nm thick on FTO, and a columnar growth was observed on AFM images (see Appendix A), as discussed previously. The size of the microstructure depends on the deposited thickness and will be the subject of another investigation.

The values of tip work function $\Phi_{\mathrm{tip}}\;$ were evaluated using a gold layer and an FTO layer. Good agreement with the literature was obtained (5.10 eV ± 0.10 eV for gold, 4.85 eV ± 0.10 eV for FTO). Perovskites work functions $\Phi_{\mathrm{perovskite}}$ were deduced from: $\Phi_{\mathrm{perovskite}}=\Phi_{\mathrm{tip}}-{\mathrm{qV}}_{\mathrm{measured}}$.

## 3. Results and Disssion

### 3.1. Electrical Characterization of CaTi_1−y_Fe_y_O_3−δ_ Perovskite Pellets

In a previous study, CaTi_1−y_Fe_y_O_3−δ_ perovskite oxide was sintered under high oxygen partial pressure (10^−6^ to 0.21 atm) showing p-type electronic conductivity [9,11]. From the Kröger–Vink formalism, the creation of an electronic hole $\left(h^{⦁}\right)$ can be linked to the following reaction at a high temperature under high oxygen partial pressure. It also noted that high oxygen partial pressure (or O_2_) leads to reduced concentrations of oxygen vacancies $\left(V_{O}^{⦁⦁}\right)$ in the perovskite structure and increased concentration of electronic holes $\left(h^{⦁}\right)$, as described in (3).
(3)$$O_{2\;(g)}^{\times \;}{+V}_{O}^{⦁⦁}↔O_{O}^{\times \;}{+2\;h}^{⦁}$$

Figure 6 shows that the Ti substitution by Fe significantly increases the electrical conductivity of the CaTi_1−y_Fe_y_O_3−δ_ perovskite. Indeed, the Ti substitution by Fe leads to the formation of oxygen vacancies $\left(V_{O}^{⦁⦁}\right)\;$ following the reaction (4), which also leads to the creation of electronic holes (as reported in reaction (3)) and consequently, the increase of the electronic conductivity of the material.
(4)$$2\;\mathrm{CaO}+{\mathrm{Fe}}_{2}O_{3}\overset{{\mathrm{CaTiO}}_{3}}{\to }2{\;\mathrm{Ca}}_{\mathrm{Ca}}^{\times \;}+2{\;\mathrm{Fe}}_{\mathrm{Ti}}^{\prime \;}{+V}_{O}^{⦁⦁\;}+{5\;O}_{O}^{\times \;}$$

In addition, the activation energy decreases versus increasing Fe from 0.40 eV (for CTF73) to 0.26 eV (for CTF37). The activation energy value is strongly linked to the structural modification of the CaTi_1−y_Fe_y_O_3−δ_ perovskite, with the atomic Fe/Ti ratio linked to the ordered oxygen vacancies in the perovskite structure [9,11].

### 3.2. Characterization of CaTi_1−y_Fe_y_O_3−δ_ Perovskite Films Deposited by PLD

#### 3.2.1. Electrical Characterizations of CaTi_1−y_Fe_y_O_3−δ_ Perovskite Films

The nature of cation substitution in perovskite structures makes it possible to manage a large range of properties. Therefore, the Fe/Ti ratio modifies the concentration of electronic holes and the mobility of electronic holes, which ultimately governs the electrical conductivity (Table 2).

In general, the electrical conductivity of thin films decreased significantly after annealing at 500 °C under air, leading to the creation of electronic holes in CTF materials, as reported previously in the reaction (3). However, the electrical conductivity of the films were still two and three orders of magnitude lower that the pellets (Table 2). This large discrepancy of electrical conductivity between pellets and the films can be linked to the specific microstructure of PLD coating: in particular, to the reduction of the associated grain boundaries and the low crystallinity of the films (see Figure 3 X-ray diffraction patterns of PLD films). In addition, there was lower variation of electrical resistivity between the three compositions because the electrical conductivity is mainly governed by the resistivity of grain boundaries. These boundaries create proper conductive paths due to the existence of defects and impurities. Moreover, the CTF55 films showed a higher electrical conductivity due to the enhancement of the associated grain boundaries in the film. Indeed, it is assumed here that the annealing treatment under air leads to increases in the crystallinity as well as the electrical conductivity of materials.

#### 3.2.2. Optical Characterization

Figure 7 presents the transmission of the three 100 nm-thick layers of oxide of deposited on ITO. After the wavelength of 500 nm, their transmission matches that of the FTO layer, showing a weak influence of oxides. Before this wavelength, the transmission drops according to the Ti/Fe ratio. Otherwise, oscillations along the spectrum due to internal optical interferences can be used to approximate the optical index ($\tilde{n}$ = 2.7 here) [16]. High reflection was monitored when the layer was deposited on sapphire (Appendix A). Due to the high differences between optical indices of air, a proportion of the light was lost in reflection. At each interface, the intensity of the reflection coefficient is related to the equation 5 where R ≈ 0 only if $n_{1}$ ≈ $n_{2}$. Nevertheless, a CTF/active layer interface should present a lower reflection due to a diminution of the index difference between the perovskite oxide ($n_{1})$ and its juxtaposed layer ($n_{2})$. Indeed, $\tilde{n}$ of 2.7 was closer to the real optical index of the active layer, such as CH_3_NH_3_PbI_3_ (between 2 and 2.5) [17], than the optical index of air (=1) on the visible range. The difference between the reflection and the transmission was equal to the absorption of the multilayer.
(5)$${R=[(n_{1}-n_{2})/\left(n_{1}+n_{2}\right)]}^{2}$$

In particular, the CTF73 transmission followed the FTO transmission fairly well, highlighting a weak influence of the oxide layer despite its large thickness, which would be reduced in a hybrid perovskite optoelectronic device. Typical thickness for the interfacial layer is between 15 and 50 nm. This reduction is particularly important to compensate for the rather high electrical resistivity and a compromise needs to be found with the energy band bending. The interfacial layer thickness needs to be sufficiently thick to twist the energy bands and block the correct carrier, but sufficiently thin to be transparent and not too resistive. This compromise will be further investigated.

The optical band gap approximated using Tauc plots (Appendix A) led to band gaps for r = 0.5 at approximately 3.2 eV, 3.4 eV, and 3.5 eV and band gaps for r = 2 at approximately 3.8 eV, 3.9 eV, and 4.0 eV for CTF37/CTF55/CTF73, respectively. The variation of stoichiometry in favor of titanium led to a modification of the distribution of carriers and defects. The resulting band tail was higher with a larger amount of titanium. The screening of the electron Coulomb potential by surrounding carriers can reduce the repulsion between valence and conduction band electrons and, among other factors, leads to a band gap reduction. Values of gap observed here can be expected to make an electron-blocking layer and a hole conductive layer together. These layers may be able to protect the active layer from UV-light due to its low optical transmission before a wavelength of 400 nm, while allowing visible light to pass through it [18].

These work functions were compared to the literature work function of ITO, FTO, PEDOT: PSS and conduction/valence bands of CH_3_NH_3_PbI_1−x_Cl_x_, CH_3_NH_3_PbBr_3_, PCBM, BCP, Ca, Al (Figure 8) [19,20,21] (AFM topography available in Appendix A). These materials are very common for perovskite solar cells or LED [22]. The CTF material work function is well located between the ITO/FTO work function and lead perovskite valence band, which make possible its use as a hole-transporting layer.

The work functions and the optical gaps of perovskite materials are listed in Table 3. Therefore, the optical properties and the work functions present close values for all the stoichiometries. The important difference lies in the electrical conductivity, in which the CTF55 configuration has a clear advantage.

## 4. Conclusions

The optoelectronic properties of CaTi_0.7_Fe_0.3_O_3−δ_, CaTi_0.5_Fe_0.5_O_3−δ_ and CaTi_0.3_Fe_0.7_O_3−δ_ perovskite oxide films deposited by the PLD technique have been investigated. The post-annealing process in air was applied on the CTF55 film, and its effect on crystal structure and microstructure properties has been realized.

It was observed that the perovskite films deposited in three different Ti and Fe stoichiometry have particular properties in terms of optical, electrical, and work function. The optimum specifications of the perovskite oxides to be used as an interlayer can be adjusted by the change in oxygen stoichiometry and the annealing. CaTi_0.5_Fe_0.5_O_3−δ_ perovskite film can be proposed as a candidate for the p-type interlayer of optoelectronic devices (solar cells, light-emitting diodes). Although the optical properties of the three compositions are not very different from each other, the CTF55 film has the excellent advantage of high electrical conductivity from the electrical properties side.

This work will pave the way for the field of p-type interlayers and provide perspective on low-cost and low-temperature processes. The next step is to investigate the interface of such oxide with hybrid perovskite [23] and finally, to integrate these perovskite oxides as an interlayer in the fully solar cell [24,25].

## Figures and Tables

**Figure 1 materials-15-06533-f001:**
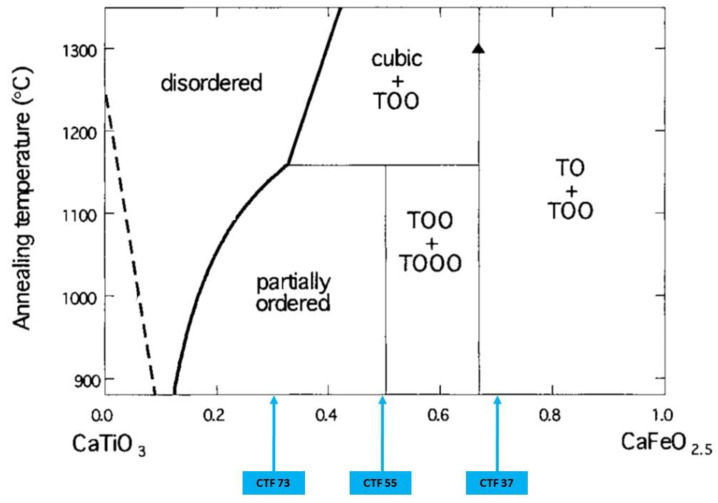
Phase diagram of the CaTiO_3_—CaFeO_2.5_ system with the three oxides explored in this study: CaTi_0.7_Fe_0.3_O_3−δ_ (CTF73), CaTi_0.5_Fe_0.5_O_3−δ_ (CTF55), and CaTi_0.3_Fe_0.7_O_3−δ_ (CTF37). The symbols TO, TOO, and TOOO correspond to the sequence of layers of tetrahedra and octahedra in the ordered structures. Reproduced with permission from [12].

**Figure 2 materials-15-06533-f002:**
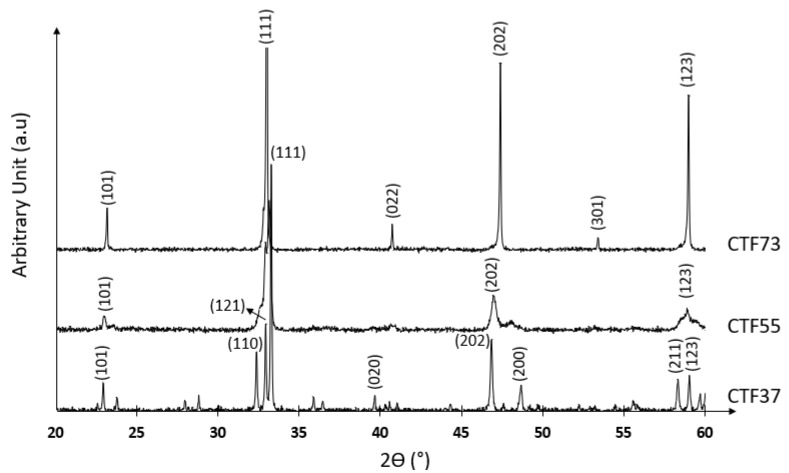
X-ray diffraction pattern of CaTi_0.7_Fe_0.3_O_3−δ_ (CTF73), CaTi_0.5_Fe_0.5_O_3−δ_ (CTF55), and CaTi_0.3_Fe_0.7_O_3−δ_ (CTF37) of pellets.

**Figure 3 materials-15-06533-f003:**
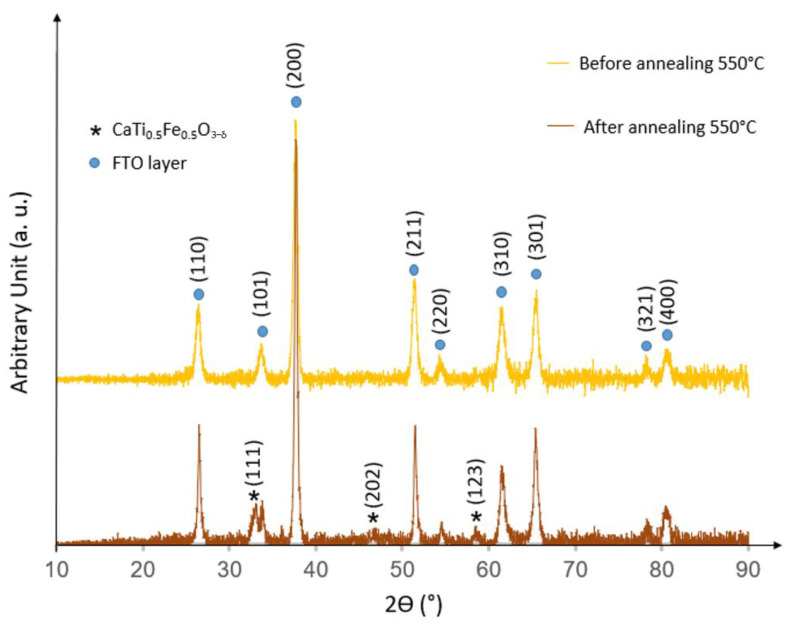
X-ray diffraction patterns of the FTO (200 nm in thickness on top of glass substrate), CTF55 perovskite films before and after annealing at 550 °C under N2 pressure.

**Figure 4 materials-15-06533-f004:**
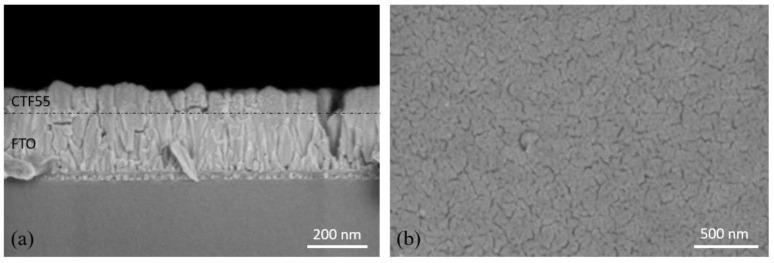
SEM images of (**a**) cross-section and (**b**) top view thin CTF55 film by pulsed laser deposition close to 100 nm in thickness on a FTO layer of 200 nm.

**Figure 5 materials-15-06533-f005:**
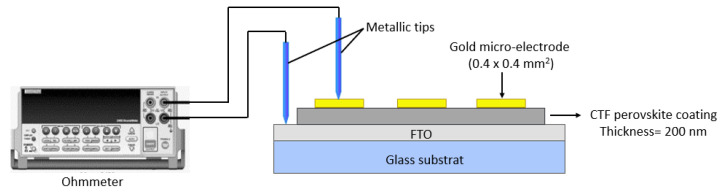
The electrical resistivity measurements of the CTF films.

**Figure 6 materials-15-06533-f006:**
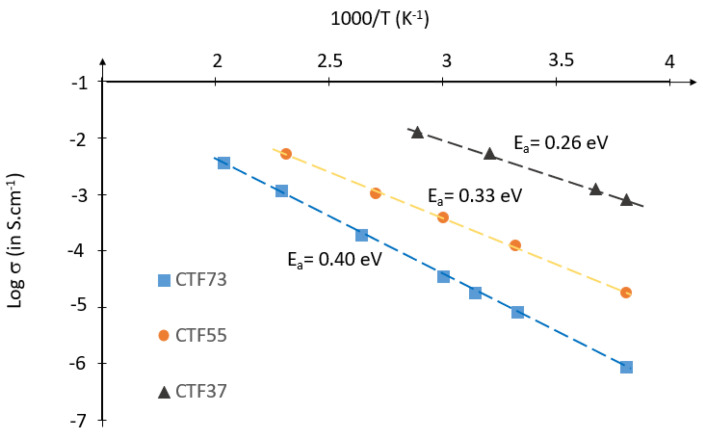
Arrhenius plot of electrical conductivity (S·cm^−1^) of CaTi_1−y_Fe_y_O_3−δ_ perovskite pellets versus 1/T (K^−1^).

**Figure 7 materials-15-06533-f007:**
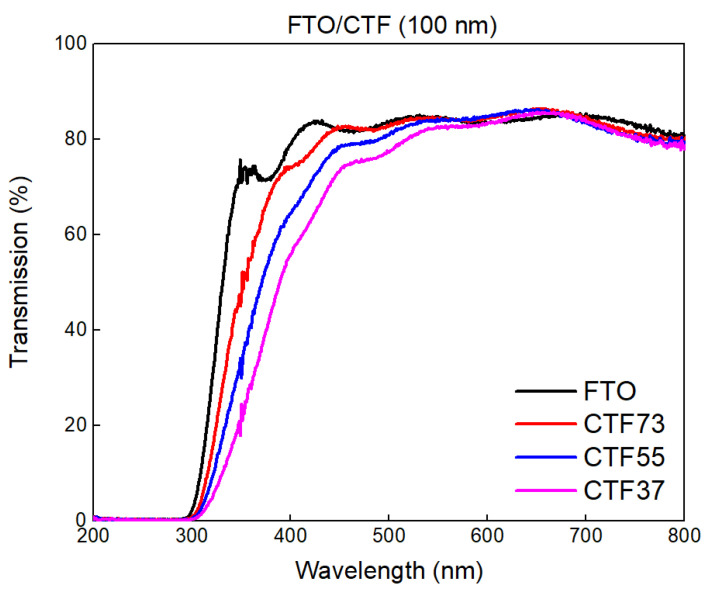
Optical properties of CaTi_0.7_Fe_0.3_O_3−δ_ (CTF73), CaTi_0.5_Fe_0.5_O_3−δ_ (CTF55), and CaTi_0.3_Fe_0.7_O_3−δ_ (CTF37), perovskite films developed by pulsed laser deposition on FTO.

**Figure 8 materials-15-06533-f008:**
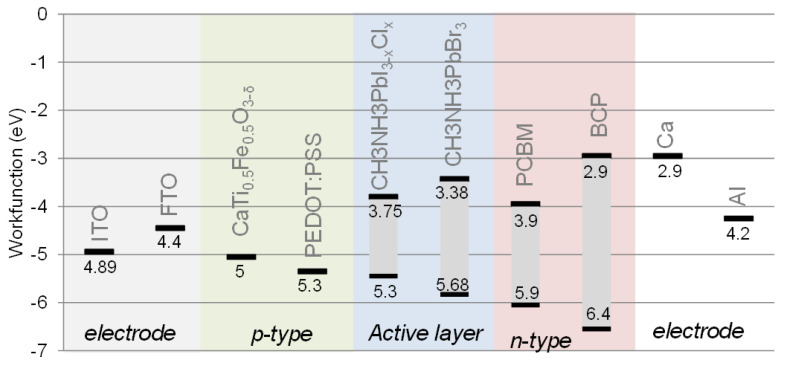
Work functions measured on CaTi_0.5_Fe_0.5_O_3−δ_ perovskites and compared to ITO, FTO, PEDOT: PSS, Ca and Al work functions and CH_3_NH_3_PbI_3−x_Cl_x_, CH_3_NH_3_PbBr_3_, PCBM, and BCP energy levels.

**Table 1 materials-15-06533-t001:** Dense CaTi_1−y_Fe_y_O_3−δ_ Perovskite Pellets Sintering Conditions.

Materials	Acronym	Sintering Conditions	Density of Starting Powders (Pycnometer, g·cm^−3^)	Relative Density of Sintered Pellets
CaTi_0.7_Fe_0.3_O_3−δ_	CTF73	1400 °C, 6 h, air	3.92	>98%
CaTi_0.5_Fe_0.5_O_3−δ_	CTF55	1400 °C, 6 h, air	3.85	>98%
CaTi_0.3_Fe_0.7_O_3−δ_	CTF37	1400 °C, 6 h, air	3.83	>98%

**Table 2 materials-15-06533-t002:** Electrical Conductivity Measurement of CTF Perovskite Materials on the Pellets and on the PLD Thin Films.

Compositions			CTF73 Perovskite	CTF55 Perovskite	CTF37 Perovskite
**Electrical** **conductivity** **(S·cm^−1^)**	Pellets		8.7·10^−7^	1.8·10^−5^	8·10^−4^
	PLD coating before annealing	Close to 200 nm (in thickness)	2·10^−10^	5·10^−10^	8·10^−10^
	PLD coating after annealing under air at 500 °C during 4 h	Close to 200 nm (in thickness)	7·10^−9^	4·10^−7^	1.3·10^−8^

**Table 3 materials-15-06533-t003:** Work Functions Measured on CaTi_1−y_Fe_y_O_3−δ_ Perovskite Films.

Materials	Acronym	Work function (eV)	Gap (eV)
CaTi_0.7_Fe_0.3_O_3−δ_	CTF73	5	3.2 (indirect)
CaTi_0.5_Fe_0.5_O_3−δ_	CTF55	5	3.4 (indirect)
CaTi_0.3_Fe_0.7_O_3−δ_	CTF37	5	3.5 (indirect)

## Data Availability

Not applicable.

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
