# Peer review of "Electrical and Optical Properties of CaTi1−yFeyO3−δ Perovskite Films as Interlayers for Optoelectronic Applications"

_materials, 2022, doi:10.3390/ma15196533_

Round 1

Reviewer 1 Report

This paper reports a research study mainly on the electric conductivity and transparency of Ca-Ti-Fe-O perovskite as a function of Ti substitution by Fe. I have following queries and concerns.

1) How do the stoichiometries of CTF73, CTF55, CTF37 as illustrated in FIG 1  are determined? It needs to be explained in more detail.

2) After being deposited into a film, the XRD spectrum of CTF55 changes considerably as compared to its pellet form (FIG 3). An explanation regarding this change is lacking.

3) Does the stoichiometry of CTF remain the same as that of the pellet after PLD deposition?

4) The symbols in Equations 3 and 4 are not well explained. The explanations of the corresponding reactions are not clear and poorly understood.

5) Why is  the conductivity of CTF55  larger than the other two CTF samples?

6) In fact, to study a trend of parameter's change in a sample system, as the case of conductivity and transparency here, only THREE samples are not enough. More samples with different Ti/Fe ratios are needed.

7) In table 2, the conductivity of 100 nm CTF37 sample decreases significantly after annealing, but those of the other two increase.  An explanation is needed for this "unusual" result. 

8) Tauc plots of absorption spectra of CTF samples are missing.

9) Why does the bandgap width increase steadily from CTF37, CTF55 to CTF73?

10) This paper is not well organized and presented, and some grammar errors exist.

Reviewer 2 Report

1. Please indicate the specific position of each peak in the XRD diffraction patterns. In addition, what about the XRD diffraction patterns of CTF73 and CTF37 films?

2. The CTF55 film is grown on the FTO/glass substrate. What is the thickness of the FTO? Please mark it in Fig. 4(a).

3. Line 125: “2.3. Electrical and Optical Characterizations”, there are only electrical characterizations here, not optical characterizations. Please re-describe it.

4. For the description of “CTF perovskite coating Thinkness = 200 and 100 nm” in Fig. 5, what are the materials with thicknesses of 200 nm and 100 nm? The expression here is not clear, please re-describe it in detail.

5. For some professional terms, please write the full name when you describe them for the first time. such as "SEM" and "AFM".

6. What about the RMS roughness for the CTF perovskite films grown on FTO glass? Please show and discuss the AFM images in the part “Surface Potential Measurements”?

7. Please check the superscript and subscript of the unit and material names in the title of Fig. 7.

8. The conductivity value of CTF55 perovskite film before and after annealing at 500℃ is constant, please explain the reason for this phenomenon.

9. Please revise your manuscript carefully to avoid any spelling and syntax errors.

Reviewer 3 Report

In this report the authors synthesized CaTi1-yFeyO3-δ perovskite thin films and studied their optical and electrical properties for optoelectronic applications, especially considering the affects of substitutions of Fe on Ti sites. The report is recommended to be accepted after minor revision, and some comments are listed below:

1. On Page 3, Figure 2, from a solid state view, "TO + TOO" or "TOO + TOOO" corresponds to a mixture of two different perovskite-related phases, with different long-range layer sequences in their structures. Especially the pXRD pattern of CTF37, it looks like a mixture of multiple phases instead of a uniform one. As it is a solid solution system, it is important to make a uniform distribution of composition. So more explanation may be added here.

2. XRD and SEM characterization results should be provided for three post-PLD-deposition films, with different substitution ratios. Theoretically the shifts of peak positions on pXRD and the morphology will be important to compare between different compositions. For SEM, EDX is recommended to be used to test the target pellets or the films, in order to confirm the uniform elemental distribution, and the rough chemical composition ratios.

3. A simple hot-probe measurement (common for semiconductor materials) on the target pellets or the deposited thin films can help to confirm the p-type feature of the CaTi1-yFeyO3-δ series.

Round 2

Reviewer 1 Report

The authors have answered all my questions and concerns, and corresponding corrections have been made in the revised manuscript. I suggest its publication.  

Reviewer 2 Report

I think that the revised manuscript is suitable for publication in the journal of Materials.